# Proline/arginine dipeptide repeat polymers derail protein folding in amyotrophic lateral sclerosis

Maria Babu [1], Filippo Favretto[1], Alain Ibáñez de Opakua[1], Marija Rankovic[2], Stefan Becker [2✉] & Markus Zweckstetter [1,2✉]

Amyotrophic lateral sclerosis and frontotemporal dementia are two neurodegenerative diseases with overlapping clinical features and the pathological hallmark of cytoplasmic deposits of misfolded proteins. The most frequent cause of familial forms of these diseases is a hexanucleotide repeat expansion in the non-coding region of the *C9ORF72* gene that is translated into dipeptide repeat polymers. Here we show that proline/arginine repeat polymers derail protein folding by sequestering molecular chaperones. We demonstrate that proline/arginine repeat polymers inhibit the folding catalyst activity of PPIA, an abundant molecular chaperone and prolyl isomerase in the brain that is altered in amyotrophic lateral sclerosis. NMR spectroscopy reveals that proline/arginine repeat polymers bind to the active site of PPIA. X-ray crystallography determines the atomic structure of a proline/arginine repeat polymer in complex with the prolyl isomerase and defines the molecular basis for the specificity of disease-associated proline/arginine polymer interactions. The combined data establish a toxic mechanism that is specific for proline/arginine dipeptide repeat polymers and leads to derailed protein homeostasis in *C9orf72*-associated neurodegenerative diseases.

[1] German Center for Neurodegenerative Diseases (DZNE), Göttingen, Germany. [2] Department for NMR-based Structural Biology, Max Planck Institute for Biophysical Chemistry, Göttingen, Germany. ✉email: sabe@nmr.mpibpc.mpg.de; Markus.Zweckstetter@dzne.de

The pathological hallmark of neurodegenerative diseases is the deposition of misfolded aggregated proteins in the cytosol of neurons. In the two neurodegenerative diseases amyotrophic lateral sclerosis (ALS) and frontotemporal dementia (FTD) the proteins transactive response DNA-binding protein 43 (TDP-43), fused in sarcoma and superoxide dismutase, aggregate in degenerating motor neurons[1]. In addition, hexanucleotide repeat expansions of the *C9orf72* gene are translated into dipeptide repeat polymers that form cytoplasmic inclusions in the brains of diseased patients[2,3]. The *C9orf72*-repeat polymers made up of glycine/arginine (GR) and proline/arginine (PR) dipeptides bind a large number of cellular proteins including molecular chaperones[4–7]. Despite clear evidence that derailed protein homeostasis is central to the pathology, the molecular mechanism that causes protein misfolding and aggregation in *C9orf72*-ALS/FTD remains unknown.

Hexanucleotide repeat expansions of the *C9orf72* gene are the most frequent cause of familial ALS and FTD as well as some sporadic cases[8,9]. The toxicity of hexanucleotide repeat expansion has been attributed to two distinct, but potentially synergistic mechanisms[4]. In the loss-of-function mechanism, neuronal toxicity is caused by a reduced expression of the C9ORF72 protein[8,10–12]. In the gain-of-toxic-function mechanism, cellular toxicity arises either directly from repetitive RNA transcribed from the hexanucleotide expansion region or from long dipeptide repeat polymers translated from the repeat RNA[2–4,13–16]. Studies on cellular and animal models demonstrated that *C9orf72* repeat polymers made up of proline/arginine dipeptides are most toxic[5,17,18].

Molecular chaperones play an important role in protein homeostasis, because they help proteins to fold[19]. *C9orf72*-associated proline/arginine repeat polymers bind in cells to molecular chaperones termed prolyl isomerases[5–7]. In addition, low levels of the prolyl isomerase PPIA in peripheral blood mononuclear cells are associated with early disease onset in ALS patients[20,21]. A link between neurodegeneration, protein misfolding, and prolyl isomerase activity is further supported by knock-out studies in mice[22]: when the gene of the prolyl isomerase PPIA is deleted, mice develop a neurodegenerative disease that recapitulates features of FTD, including the aggregation of TDP-43 into cytoplasmic deposits[22].

Here we provide insight into the molecular mechanism of impaired chaperone activity and derailed protein homeostasis by PR repeat polymers in *C9orf72*-associated neurodegenerative diseases.

## Results

**PR repeat polymers inhibit prolyl isomerase folding activity.** *C9orf72* repeat polymers made up of PR and GR dipeptides interact with a large number of proteins in the cell[5–7,23–30]. Comparison of the corresponding interactomes demonstrated that 65 proteins interact with PR but not GR repeat polymers[5]. Gene ontology analysis further showed that a class of PR-specific interactors are prolyl isomerases (Fig. 1a). The identified prolyl isomerases include PPIA, PPIB, and PPIF[5]. PPIA was independently identified as an interactor of PR repeat polymers in two other interactome studies[6,7]. Knock-out of PPIA causes aggregation of TDP-43 in mice and neurodegeneration[22].

To gain insight into the consequences of the aberrant interaction of PR repeat polymers with prolyl isomerases, we performed protein folding assays. We unfolded the ribonuclease RNaseT1 in urea and diluted the denaturant to trigger refolding[31]. Upon dilution, RNaseT1 refolding started immediately and was completed within ~30 min (Fig. 1b and Supplementary Fig. 1)[31]. We then refolded RNaseT1 in the presence of a large excess of a PR repeat polymer with 20 repeats (PR20). The refolding kinetics of RNaseT1 were unchanged in the presence of PR20 (Fig. 1b and Supplementary Fig. 1).

Next, we performed refolding assays in the presence of the prolyl isomerase PPIA. In agreement with the ability of PPIA to catalyze the *cis–trans* isomerization of prolyl bonds[19,31], addition of PPIA strongly accelerated the folding of RNaseT1 (Fig. 1b and Supplementary Fig. 1). However, increasing concentrations of PR20 inhibited the catalyzing effect of PPIA in a dose-dependent manner (Fig. 1b, c and Supplementary Fig. 1). The concentrations of PR20 to achieve inhibition are larger than those at which the immunosuppressant drug cylcosporin A inhibits the PPIA-catalyzed refolding of RNaseT1 (refs. [19,31]), which is in agreement with the involved affinities: cyclosporin A binds three orders of magnitude stronger to the enzyme ($13 \pm 4$ nM[32]) when compared to the PR repeat polymer (~10–80 µM; see below). The combined data show that *C9orf72*-associated PR repeat polymers do not affect the self-folding process of proteins, but selectively inhibit the folding catalyst activity of PPIA.

**PR repeat polymers bind to the active site of PPIA.** To gain molecular insight into the PR-mediated inhibition of enzyme-catalyzed folding, we analyzed the binding of PR repeat polymers to PPIA using NMR spectroscopy. Upon addition of increasing concentrations of the 20-repeat PR polymer PR20, the NMR signals of PPIA broadened in a dose-dependent manner (Fig. 2a and Supplementary Fig. 2a, b). Because no new cross-peaks appeared in the NMR spectrum of PPIA upon addition of PR20, the exchange between the unbound and PR20-bound state of PPIA is in an intermediate to slow exchange regime.

To identify the binding site of PR20 in PPIA, we analyzed the PR20-induced signal broadening for the individual residues of PPIA. The analysis located the strongest intensity decreases within or next to the enzyme's active site (Fig. 2b, c). The perturbed PPIA residues include Arg55, Gln111, Asn102, and Glu120. Arg55 has been identified in previous X-ray studies of PPIA in complex with Xaa-proline dipeptides where this residue is shown to make hydrogen bonds to the proline carbonyl[33]. Gln111, on the other hand, is not making direct contacts with the PR20 chain, but is affected by mutation of Arg55 and is part of a dynamic network of residues in the binding pocket[34].

In contrast to the PR repeat polymer, repeat polymers made up of either alanine/proline (AP) or glycine/proline (GP) dipeptides displayed less signal broadening and more chemical shift changes (Fig. 2a and Supplementary Fig. 2c–f). The binding processes are thus in the intermediate/fast for AP20, and in the fast exchange regime for GP20. No or very little broadening or shifts of the NMR signals of PPIA were observed when an eightfold excess of the 20-dipeptide polymer GR20 was added to PPIA (Fig. 2a, b). The glycine/arginine repeat polymer thus does not bind to PPIA.

Next, we determined the PPIA affinity of the PR repeat polymer using complementary methods. Fitting the NMR broadening data of the catalytic residue Arg55 for increasing PR polymer concentrations resulted in the dissociation constant $23 \pm 7$ µM. In addition, we quantified the affinity using isothermal titration calorimetry (Fig. 2d). The calorimetry-derived $K_d$ value is ~50 µM. Notably, variations in $K_d$ values were previously observed for PPIA interactions when different methods were used[35,36]. The NMR/calorimetry-derived micromolar affinity of the PR repeat polymer to PPIA is comparable to that of other PPIA/protein interactions[35,36]. In contrast, the PPIA affinities of the dipeptide repeat polymers AP20 ($K_d = 597 \pm 12$ µM) and GP20 ($K_d = 1188 \pm 64$ µM) are more than one order of magnitude weaker (Supplementary Fig. 2d, f). Because GP20 and AP20 have the same number of proline residues as PR20, the analysis demonstrates that both proline and arginine residues are important for efficient binding of *C9orf72*-repeat polymers to prolyl isomerases.

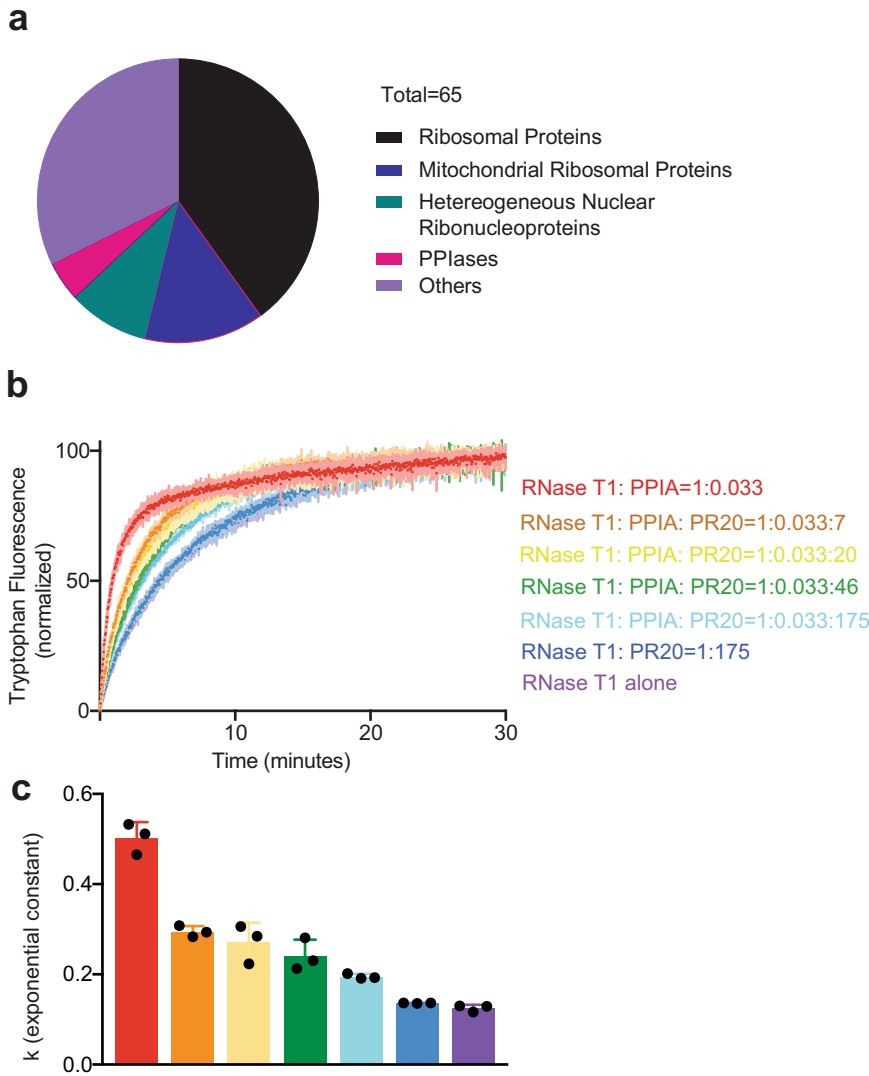

**Fig. 1 C9orf72-associated PR repeat polymers inhibit prolyl isomerase folding activity. a** Classification of 65 PR polymer-specific-binding proteins. The data for the analysis were taken from the list of dipeptide repeat polymer interactors identified by Lee et. al. (Table S1 in ref. [5]). Only those interactions with a saint score[48] more than 0.9 were included. **b** Inhibition by the 20-dipeptide polymer PR20 of the catalytic effect of PPIA on protein folding of RNaseT1. The increase in fluorescence at 320 nm is shown as a function of the time of refolding in the absence of PPIA and PR20 (violet, "RNaseT1 alone"), and in the presence of a fixed concentration of PPIA and increasing concentrations of PR20 (red, orange, yellow, green, and light blue represent 0, 7, 20, 46, and 175 times excess of PR20 with respect to RNaseT1, respectively). The control experiment showing the refolding of RNaseT1 when PPIA is not present but PR20 has been added is also displayed (dark blue). **c** Histogram shows the mean value of the exponential folding rate constants $k$ of RNaseT1 in the absence and presence of PPIA and increasing PR20 concentrations. Color coding as in **b**. $n = 3$ independent experiments were performed. Error bars represent standard deviation from mean value. $k$ values of each independent experiment are shown in black dots for every condition.

To gain insight into the critical length of PR repeat polymers for binding to prolyl isomerases, PPIA was titrated with PR polymers of decreasing repeat number. Repeat polymers with ten (PR10) and five (PR5) PR dipeptides efficiently bound to PPIA (Supplementary Fig. 2a). For PR10 we determined the dissociation constant $27 \pm 5\,\mu M$ (Fig. 2e), i.e. slightly lower than the PR20 affinity. In the case of PR5, the $K_d$ value further decreased to $49 \pm 16\,\mu M$ (Fig. 2e). Additional shortening of the peptide chain to two PR dipeptides abrogated the binding to PPIA: even an eightfold excess of PR2 did not influence the enzyme's NMR signals (Supplementary Fig. 2a).

**Structure of PR repeat polymer in complex with PPIA.** To reveal the structural basis of derailed protein homeostasis by PR repeat polymers, we determined the crystal structure of a PR repeat

polymer in complex with the prolyl isomerase PPIA. The structure of the PR20/PPIA complex was resolved at 1.3 Å resolution (Supplementary Tables 1 and 2). In the heterodimeric complex, four PR repeats are in direct contact with the catalytic pocket of the enzyme (Fig. 3a). The functionally important PPIA-residue Arg55 forms hydrogen bonds with the carbonyl group of a proline–arginine peptide bond (Fig. 3b and Supplementary Fig. 3). The proline residue in this specific peptide bond has a *cis* conformation, suggesting isomerase activity of PPIA on the PR repeat polymer. In addition, several other intermolecular contacts are present in the structure of the complex: Trp121 forms a hydrogen bond with the carbonyl group of the adjacent arginine–proline peptide bond; Asn102 and Glu120 of PPIA form hydrogen bonds with the arginine side chains of the PR repeat polymer (Fig. 3b).

The crystal structure of the PR repeat polymer bound to the prolyl isomerase is unique when compared to known protein/

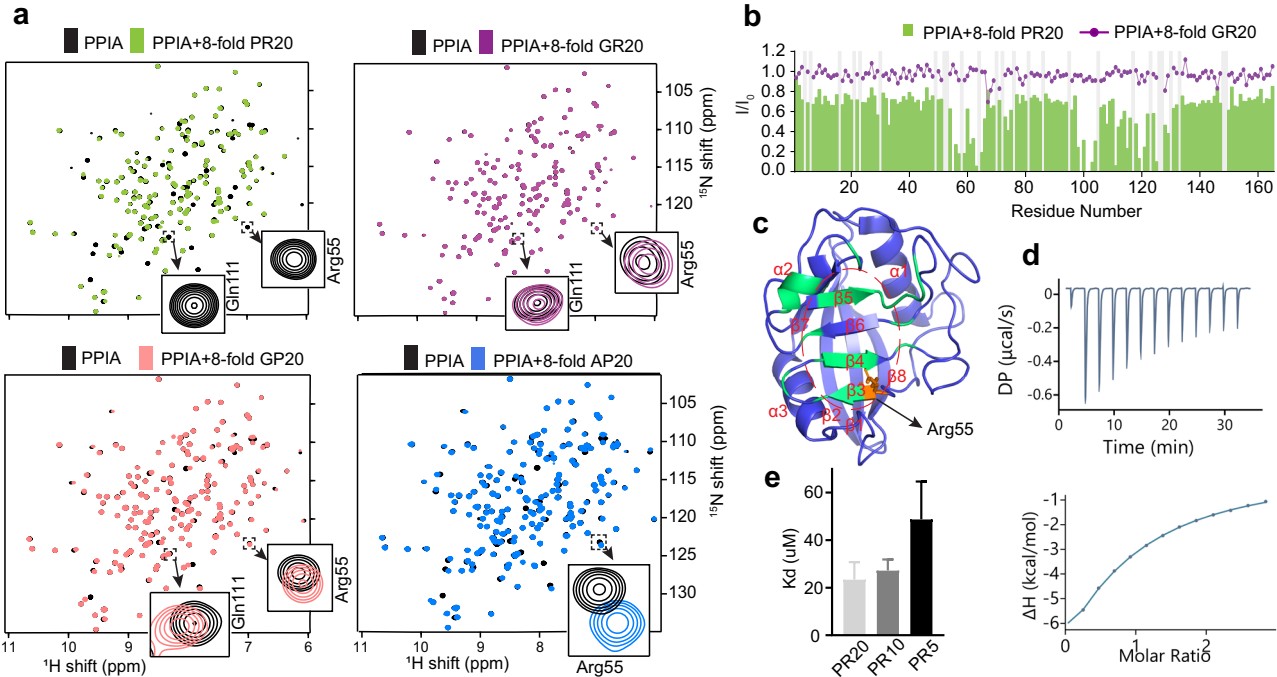

**Fig. 2 PR repeat polymers bind to the active site of PPIA. a** [1]H–[15]N HSQC spectra of PPIA alone (black) and in the presence of an eightfold excess of PR20 (green), GR20 (magenta), GP20 (red), and AP20 (blue). Cross-peaks of residues, which are located in the substrate-binding pocket of PPIA, are highlighted. **b** Changes in the intensities of HSQC peaks of PPIA upon addition of an eightfold excess of PR20 (bars). $I$ and $I_0$ are the intensities of the PPIA cross-peaks in the presence and absence of PR20. No broadening was observed upon addition of an eightfold excess of GR20 (line). **c** PPIA residues with strong PR20-induced signal attenuation are highlighted in the 3D structure of PPIA (PDB code: 5KUZ; https://www.wwpdb.org/pdb?id=pdb_00005kuz). Arg55 is highlighted in orange. PPIA residues with $I/I_0$ values less than 0.394 (mean value of $I/I_0$ for all residues minus its standard deviation) upon addition of eightfold molar excess of PR20 are shown in green. The active site of PPIA is circled (dashed line). **d** Isothermal titration calorimetry thermogram of PR20 binding to PPIA. **e** $K_d$ values for the interaction of PPIA with PR20, PR10, and PR5 derived from the attenuation of the HSQC cross-peak of Arg55 of PPIA. Error bars represent the standard deviation in $K_d$ generated from least-square fitting of experimental data to Eq. (1).

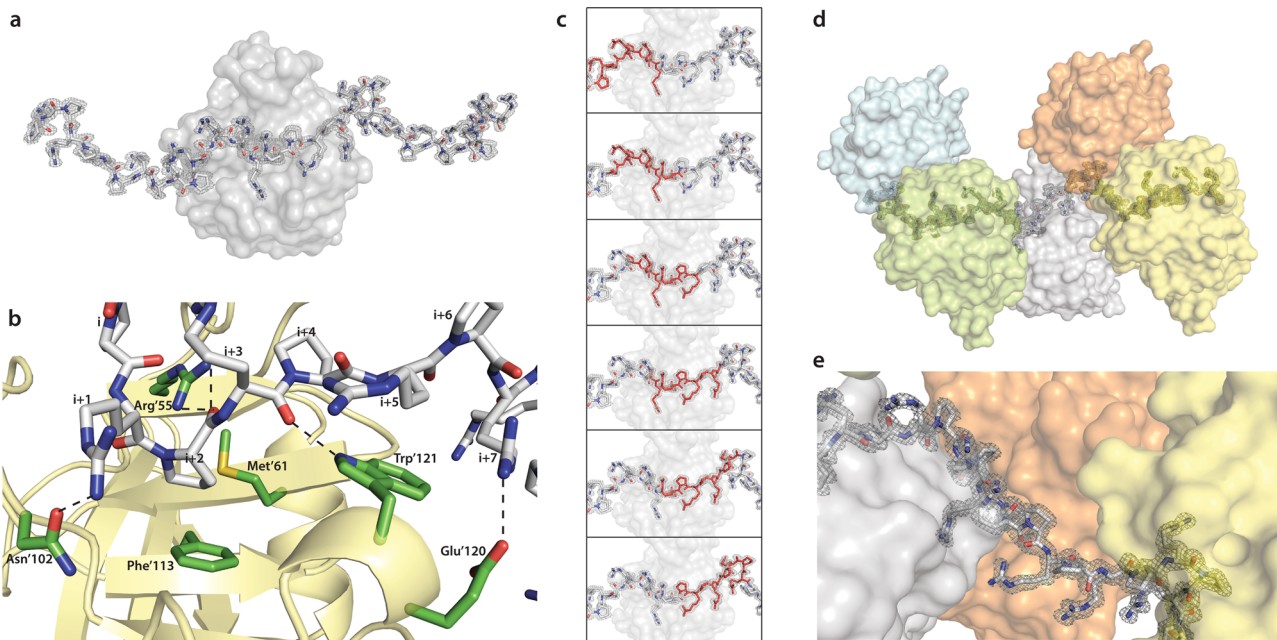

**Fig. 3 Structural basis of chaperone inhibition by PR repeat polymers. a** PR repeat polymer in complex with the ALS/FTD-associated prolyl isomerase PPIA. Only eight residues of the PR polymer could be built into the asymmetric unit; the displayed polymer chain was assembled from symmetry mates (2 $mFo–DFc$ electron density map of PR20 contoured at 1.4σ level, depicted in gray). **b** Close-up view of the interface between the PR repeat polymer (gray) and PPIA (yellow/green). Hydrogen bonds are depicted by dashed black lines. **c** The continuous electron density for the PR polymer throughout the crystal lattice allowed a slider-like positioning of its N terminus. **d, e** Selected regions from the crystal lattice displaying continuous electron density of the PR polymer chain (gray stick model with semi-transparent electron density). PPIA molecules are shown in surface representation.

protein complex structures (Fig. 3c–e). The chaperone-bound PR polymer chain displays a continuous electron density without interruption throughout the crystal lattice (Fig. 3c–e). However, the electron density of a single asymmetric crystal unit can only accommodate four PR dipeptides. Always four PR dipeptides from the polymer bind in an identical manner (residues I to I + 7 in Fig. 3b) to the respective PPIA molecule in each asymmetric unit (Fig. 3d, e). Due to the continuous electron density of the PR chain in the crystal lattice, the PR repeats are randomly positioned, i.e. no unique start of the polymer chain is present (Fig. 3c). This suggests that the chaperone can bind to any of the PR dipeptides in the repeat polymer and the electron density is a result of this averaging process. The continuous electron density thus provides an atomic-resolution view of a large number of chaperone molecules bound to a long PR repeat polymer (Fig. 3d).

## Discussion

While the link between *C9ORF72* hexanucleotide repeat expansion and ALS/FTD has been firmly established, diverse molecular mechanisms including perturbation of membrane-less organelles and nucleocytoplasmic transport, amyloid formation of GA repeat polymers, as well as translation repression have been suggested to drive neuronal dysfunction[4–7,13,23–30]. Dipeptide repeat polymer-associated toxicity is likely mediated by aberrant protein/protein and protein/RNA interactions, in particular of the arginine side chains of PR and GR repeat polymers[5–7,23–30]. However, PR repeat polymers were found to be more toxic than GR repeat polymers in cell and animal models of *C9ORF72*-ALS/FTD[5,17,18]. This indicates that the toxic mechanism of PR repeat polymers cannot be fully attributed to arginine residues, but depends on the unique combination of arginine with proline in PR repeat polymers.

PPIA is a major cellular chaperone that catalyzes *cis/trans* isomerization of prolyl peptide bonds and thus helps proteins to fold when they exit from the ribosome (Fig. 4a)[19]. Our experiments show that PR repeat polymers inhibit the folding catalyst activity of PPIA, thereby contributing to the disruption of protein homeostasis (Fig. 4a)[37]. When we then compare the PPIA/PR complex structure with the structure of PPIA in complex with the natural immunosuppressant cyclosporin A[38], we find a similar inhibition mechanism: in both complexes the side chain of Arg55, which is critical for catalysis of PPIA-mediated *cis*-/*trans* isomerization[39], forms strong contacts with the molecule that inhibits PPIA's folding catalyst activity (Fig. 4b). The atomic structure of the PR repeat polymer in complex with PPIA defines the contribution of both the arginine and the proline residues of PR repeat polymers to protein/protein interactions at high resolution and thus provides a potential starting point for the development of molecules that specifically bind to PR repeat polymers.

We found that short PR repeat proteins bind with micromolar affinity to the prolyl isomerase PPIA. This affinity is comparable to values reported for other protein substrates[35,36], but lower than the nanomolar affinity of the drug cyclosporin[32]. However, long PR polymers can potentially bind many chaperones simultaneously: based on the crystal structure of the PPIA/PR20 complex a single PR polymer with a repeat size of 400 can sequester up to 100 prolyl isomerase molecules. In addition, the molecular mechanism changes from a simple one-site binding process to the binding to a one-dimensional lattice with a huge number of potential binding sites (Fig. 4c)[40]. Aberrant interactions of *C9orf72*-repeat polymers might thus be mechanistically similar to the binding of transcription factors to DNA in which transcription factors bind to the one-dimensional lattice of binding sites in the DNA[40]. When PPIA molecules bind to proximal sites in the

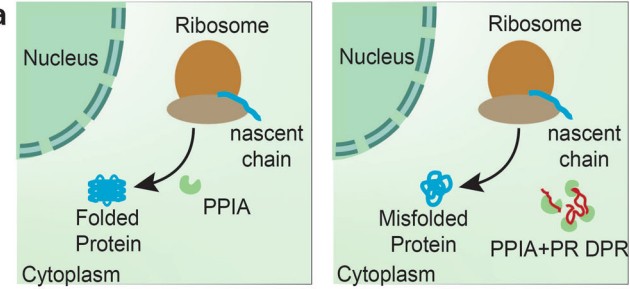

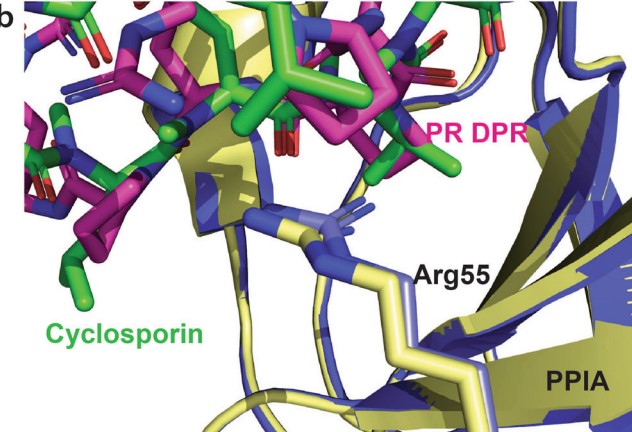

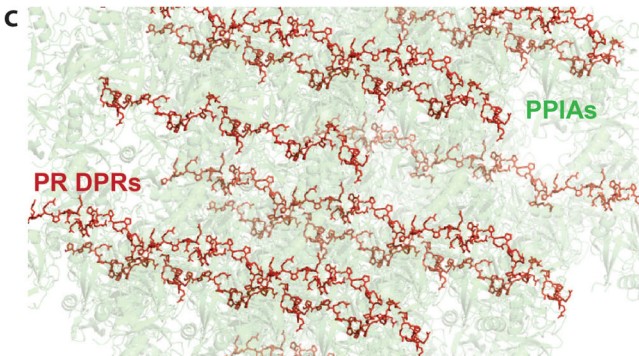

**Fig. 4 Inhibition and sequestration of prolyl isomerases by *C9ORF72* ALS/FTD-associated PR repeat polymers. a** In normal conditions not associated with disease, prolyl isomerases such as PPIA catalyze the *cis–trans* isomerization of prolyl peptide bonds and thus help proteins to fold when they exit from the ribosome. PR dipeptide repeat polymers (DPR; red chains) translated from hexanucleotide repeat expansion in the non-coding region of the *C9ORF72* gene, bind to the active site of prolyl isomerases and thus block their ability to catalyze protein folding. **b** Comparison of the active site of the prolyl isomerase PPIA (blue/yellow) in complex with the immunosuppressant drug cyclosporin A (green; PDB code: 1CWA; https://www.wwpdb.org/pdb?id=pdb_00001cwa[38]) and a PR repeat polymer (pink). **c** Crystal lattice of the PPIA/PR20 complex illustrating the dense packing of PPIases (green) that is possible on long PR dipeptide repeat polymers (DPR, red).

effectively one-dimensional lattice of the PR repeat chain, direct interactions between PPIA molecules might further contribute to avidity in the inhibition of PPIA by PR dipeptide repeat polymers. Taken together our study provides a molecular mechanism for the derailment of protein homeostasis by PR repeat polymers in *C9orf72*-associated neurodegenerative diseases. In addition, it warrants further studies to investigate the interaction of PR repeat polymers with PPIA in neurons including their subcellular localization.

## Methods

**Protein preparation**. The gene of human PPIA was cloned into a modified pET28a vector (Addgene). The PPIA plasmid was transformed in *Escherichia coli* BL21 (DE3) cells (Novagen). Cells were cultured in LB medium at 37 °C. When the OD value reached 0.6, protein overexpression was induced with 0.4 mM IPTG. After 15 h of incubation at 16 °C, cells were centrifuged down and resuspended in the resuspension buffer (20 mM sodium phosphate, 300 mM NaCl, 2 mm DTT, 5 mM imidazole, 0.01% NaN3, pH 7.2) additionally supplemented with 100 mM PMSF and 100 mg/ml lysozyme at pH 7.2. The cells in this buffer were sonicated, centrifuged, and the supernatant was loaded onto a Ni-NTA Agarose column. The column was washed with resuspension buffer containing 10 and 25 mM imidazole; protein was eluted with the resuspension buffer supplemented with 300 mM imidazole. The His-tag was cleaved by incubating the protein with TEV protease in a dialysis bag (MWKO 3.5 kD) at room temperature overnight while dialysing it to His-tag cleavage buffer (20 mM sodium phosphate, 150 mM NaCl, 2 mM DTT, 0.1 mM EDTA at pH 7.7). Next, the protein solution was again loaded onto the 5 ml Ni-NTA Agarose column. The His-tag-cleaved PPIA came out in the flow through. It was concentrated and loaded onto a gel filtration column (Superdex 75, GE Healthcare Life Sciences) and eluted with 50 mM HEPES, 150 mM NaCl, 3 mM DTT, 0.02% NaN3, pH 7.4. Finally, the protein was dialyzed into the respective buffers for protein folding assays, NMR experiments, and crystallography as specified below.

$^{15}$N-labeled PPIA was produced by culturing and inducing the cells in M9 minimal media supplemented with $^{15}$NH$_4$Cl (Cambridge isotope Laboratories).

PR20, GR20, GP20, AP20, PR10, PR5, and PR2 peptides were synthesized by solid-phase peptide synthesis. All peptides were acetylated and amidated at the N- and C-termini, respectively. Peptide stocks were prepared by weighing and dissolving the required amount of powder in the buffers used for protein folding assays and NMR spectroscopy as specified below.

**Protein folding assay**. RNaseT1 (purchased from Thermo Fischer) was unfolded by incubating in 6.9 M urea at 10 °C for 2 h in 100 mM Tris-HCl buffer at pH 8. Refolding of RNaseT1 was initiated by diluting it 35 times with same buffer such that the final concentration of RNaseT1 was 2.27 μM and urea 0.197 M. Tryptophan fluorescence emission was measured at 320 nm (excitation wavelength = 280 nm) during refolding for 1 h on a Cary Eclipse Fluorescence Spectrophotometer at 10 °C. In experiments with PR20, PPIA, or both, they were added to the dilution buffer and incubated at 10 °C for 2 h prior to mixing. Data were normalized and averaged for graphical representation. The exponential constants, *k*, for the increasing fluorescence intensities were obtained from fitting a mono-exponential function to the experimental data in Graph Pad Prism.

**NMR spectroscopy**. Titration of $^{15}$N-labeled PPIA with dipeptide repeat polymers (PR20, GR20, GP20, AP20, PR10, PR5, and PR2) were carried out in 25 mM HEPES buffer with 100 mM NaCl, 2 mM DTT, 0.01% NaN3, and 10% D20 at pH 7.4. For 40 μM protein, the ligand concentrations were increased from 0 to 320 μM in five steps. For the PPIA/GP20 and PPIA/AP20 titrations, the concentration of the repeat polymer was further increased up to 3200 μM. Two-dimensional $^{15}$N-$^1$H HSQC spectra[41] were acquired at 298 K for each PPIA/repeat polymer ratio. The PPIA/GP20 and PPIA/PR2 titrations were performed on a 700 MHz spectrometer (Bruker), whereas all others were performed on a 800 MHz spectrometer (Bruker), both equipped with cryoprobes. The spectra were processed with Topspin 3.6.2 (Bruker) and analyzed with CCPNmr 2.4.2 (ref. [42]).

$K_d$ calculations for the interactions of PPIA with PR repeats (PR5, PR10, and PR20) were made from the dipeptide repeat polymer-induced variations in PPIA cross-peak intensities, using the equation for slow exchange regime interactions:

$$1 - \frac{I}{I_0} = I_{max} \left[ \frac{(K_d + P + x) - \sqrt{(K_d + P + x)^2 - 4 \times P \times x}}{2P} \right] \quad (1)$$

where *I* is the intensity of PPIA peaks in the presence of increasing ligand concentrations, $I_0$ is the intensity of free protein, *P* is the total concentration of protein, *x* is the concentration of ligand, and $K_d$ is the dissociation constant.

The error in $1-(I/I_0)$ is taken to be the error in the experimental quantity of $I/I_0$ and has been calculated from the noise in HSQC spectra according to

$$\sigma_{(I/I_0)} = \left( \frac{I}{I_0} \right) \sqrt{ \left( \frac{\sigma_{(I_0)}}{I_0} \right)^2 + \left( \frac{\sigma_{(I)}}{I} \right)^2 } \quad (2)$$

where $\sigma(I/I_o)$ is the error in $I/I_0$, $\sigma_{(I_o)}$ and $\sigma_{(I)}$ are the noise (RMS value of background noise from various regions) in the HSQC spectrum of PPIA alone and that of PPIA in the presence of ligand, respectively.

In case of the PPIA/GP20- and PPIA/AP20 interactions, the $K_d$ value was calculated from the equation for fast exchange interactions using chemical shift perturbations (CSP) according to

$$CSP = CSP_{max} \left[ \frac{(K_d + P + x) - \sqrt{(K_d + P + x)^2 - 4 \times P \times x}}{2P} \right] \quad (3)$$

where

$$CSP = \sqrt{(\delta_H)^2 + \left( \frac{\delta_N}{6.5} \right)^2} \quad (4)$$

$I_{max}$ and $CSP_{max}$ were treated as independent free fit parameters for the residues. The error value in the $K_d$ is the standard error of fitting.

**Isothermal calorimetry**. PPIA and PR20 stocks were prepared in 25 mM HEPES, 100 mM NaCl, 1 mM TCEP at pH 7.4. Forty-four micromolar PPIA in the cell was titrated with 650 μM PR20 in the syringe. The titration was performed at 10 °C employing 13 steps of injection. PPIA/buffer, buffer/PR20, and buffer/buffer titrations were also performed and assigned as controls for the analysis of the binding curve. Experiments were performed with a Malvern Microcal PEAQ-ITC instrument. During fitting the number of binding sites was fixed to one.

**X-ray crystallography**. For crystallization, the PR20 peptide was added in fivefold molar excess to PPIA. The total protein concentration was adjusted in 25 mM HEPES, 2 mM DTT, 0.01 % NaN$_3$ at pH 7.4 to 20 mg/ml. Crystals were obtained at 20 °C by sitting drop vapor diffusion using 0.1 M HEPES, pH 7.5, 25% PEG1000 as reservoir solution. For data collection, crystals were cryoprotected in reservoir solution supplemented with 10% glycerol. Data collection was performed at 100 K at SLS Villigen, Switzerland (beamline PXII, Eiger2 16M detector, Dectris). Data were processed with XDS[43]. Space group determination and statistical analysis were performed with XPREP (Bruker AXS, Madison, Wisconsin, USA). The structure was solved by molecular replacement with PHASER[44] using the crystal structure of PPIA (PDB code: 5KUL; https://www.wwpdb.org/pdb?id=pdb_00005kul[45]) as search model. Refinement was performed with Refmac[46] alternating with manual model building in Coot[47].

**Reporting summary**. Further information on research design is available in the Nature Research Reporting Summary linked to this article.

## Data availability

Coordinates of the PR20/PPIA complex were deposited to PDB (accession code 7ABT). All other relevant data are available from the corresponding authors. Source data are provided with this paper.

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

## Acknowledgements

We thank Kerstin Overkamp for peptide synthesis, and Birger Dittrich, Tim Grüne, and George Sheldrick for useful discussions. We thank the beamline staff at SLS, X10SA for support with X-ray data collection. M.Z. was supported by the Deutsche For-schungsgemeinschaft (DFG, German Research Foundation: SFB 860/B02) and the Eur-opean Research Council (ERC) under the EU Horizon 2020 research and innovation program (grant agreement no. 787679).

## Author contributions

M.B. prepared protein, conducted folding assays, ITC and NMR experiments, as well as data acquisition and analysis; F.F. performed ITC, prepared protein, and supervised protein preparation as well as NMR experiments; A.I.d.O. and M.R. helped in experi-mental design; S.B. performed crystallization and determined the structure of the PPIA/PR20 complex; M.B., S.B., and M.Z. wrote the paper. M.Z. designed and supervised the project.

## Funding

## Competing interests

The authors declare no competing interests.
