## [Peer Review File · Nature Communications]

REVIEWER COMMENTS

Reviewer #1 (Remarks to the Author):

The authors present structural evidence of the interaction between a 20-mer repeat of a dipeptide Pro-Arg (PR) and the protein CYPB/PPIA. PR repeats are the result of aberrant protein expression that will lead to neurotoxicity in the brain and contribute to neurodegenerative diseases. Their findings would have implications in finding a molecular mechanism of how a hexanucleotide repeat expansion of the non-coding region of the C9ORF72 gene leads to Amyotrophic Lateral Sclerosis (ALS) and Frontotemporal Dementia (FTD).

The manuscript is centered in finding the binding site of the PR 20-mer on the structure of PPIA by means of NMR spectroscopy and X-ray crystallography and provides evidence of binding affinity by NMR data and a droplet disaggregation fluorescence assay. Comparison to a low binding Gly-Arg (GR) 20-mer that belongs to proteins that do not interact with PPIA is provided.

The structural evidence presented in the manuscript is sound and provides adequate basis for the conclusions drawn by the authors. The references are appropriate. However, major issues were raised:

1. The authors present two residues, Arg55 and Gln111, from their NMR studies (Figure 1b and 1c) as the candidates for binding the 20-mer PR peptide. Whereas Arg55 was confirmed by X-ray diffraction analysis of the complex between the 20-mer and PPIA, no mention of Gln111 is found neither in the main text nor in Figure 2b (which shows hydrogen bonding interactions for Arg55 only). This residue is omitted from Figure 2b. Could the authors explain why this may be happening? Why is the observation from NMR different from X-ray? In fact, Arg55 has been identified in previous X-ray studies of PPIA in complex with Xaa-Pro dipeptides (conf. Zhao and Ke, *Biochemistry* (1996), 35:7362-7368, and PDB structures 2CYH, 3CYH, 4CYH, and 5CYH) where this residue is shown to make hydrogen bonds to the Pro carbonyl while Gln111 is furthest away, not contacting the dipeptide. Several PPIA-bound structures at high resolution have been published that show similar binding interactions. The authors are encouraged to include in their manuscript a reference to these earlier works and compare their new structure to relevant peptide-bound PPIA structures.
2. The authors should clarify which parts of the 20-mer are making interactions in describing the packing of PPIA and the 20-mer. In Figure 2d, it is evident that an 8-mer in one asymmetric unit is making interactions with one face of the protein (yellow and light green), and similarly to a symmetry-related molecule (grey). However, it's not evident that there's any interaction with the other packed molecules (orange and sky-blue). In other words, one 8-mer is symmetry-related to a second 8-mer leaving a 4-mer doing a different kind of interactions (or none). Could the authors clarify and expand the discussion on this?
3. Binding measurements on the peptides and PPIA are based on NMR spectroscopy. The PR peptide shows a low micromolar binding affinity which is ~2+ orders of magnitude stronger than the GR peptide. Did the authors make measurements using a different technique, e.g. ITC, MST, or SPR? These techniques could be rapidly accessible to researchers who want to replicate the experiment (compared to having to produce labelled protein for NMR) and they would provide a direct measurement of binding affinity and stoichiometry. Would the stoichiometry from crystal packing be the same as in solution? Could you explain?
4. Minor issues
 - a. Abstract: GR abbreviation not introduced
 - b. Pg. 4: "saint score", please add reference: Choi et al., (2011) *Nat. Methods* 8: 70-73

Silvia Russi

Reviewer #2 (Remarks to the Author):

Molecular mechanism of PPIA inhibition by C9orf72-ALS/FTD associated PR DRP proteins
Babu et al.

The manuscript submitted by Babu et al. sets out to define a novel mechanism of C9orf72 repeat expansion toxicity. In this work, the authors focus on the interaction between polyPR dipeptides and peptidyl proline isomerases (specifically PPIA). The work is motivated by the observation that PPIs have been observed in the interactome of polyPR and not polyGR peptides. Further, knockout of PPIA in mouse models shows similar phenotypes to FTD. Therefore, the authors posit that polyPR toxicity might arise in part by inhibition of PPIA. To test this hypothesis, the authors demonstrate that, among C9orf72 related dipeptide repeats, polyPR has a particularly high affinity and specific interaction with PPIA.

The data presented in the manuscript provides a clear picture of polyPR binding to PPIA. This is accomplished via a combination of NMR spectroscopy to map the binding pocket on PPIA and calculate dissociation constant and X-ray crystallography to show the molecular details of the bound structure. Importantly, the authors demonstrate the PR binds substantially tighter than either the proline-free GR or the arginine-free GP dipeptides. The crystal structure outlines a series of interactions involving arginine that explain this specificity. Taken together, I have no issues with the conclusion that polyPR binds specifically to PPIA. From a structural standpoint, these results are novel. However, the fact that PR (and not other dipeptides) bind to PPIA is perhaps unsurprising given the existing information about C9orf72 interactomes and our knowledge of substrate binding to PPIs. In the interest of clarity, the authors might want to expand on what is meant by slow versus fast exchange when describing the PR versus GP samples. The PR samples look like they are in an intermediate exchange regime due to the lack of peaks associated with the bound state. I would also like the results from figure 2c/d explained in a bit more detail. It seems to me that the authors are stating that in the crystallization conditions, PPIA binds to polyPR such that one PPI abuts the next. Is this an artifact of crystallization, or is this the presumed functional bound structure? An additional set of binding experiments that can provide independent validation of the K_d and, ideally, stoichiometry would help these conclusions. Finally, the K_d is calculated as a global fit to four residues in the binding pocket. Does this mean that all 8 residues from polyPR that associate with PPIA bind cooperatively and have the same affinity?

The authors additionally characterize the catalytic efficiency of PPIA on polyPR. The NMR assay of exchange is quite interesting. I have no major concerns about this data, but I am intrigued by the apparent slower exchange rate at higher molar ratios. This effect does not seem to be explored in much detail in the text. I would like to see some further explanation for these results. It is also not clear to me where these results fit with the larger story. Seemingly the rate of cis/trans isomerization is unrelated to PR toxicity? The fact that PPIA is a functional proline isomerase for polyPR seems like a red herring. If I am off base on this, the data should be featured in the discussion and related to the rest of the manuscript.

The data connecting PPIA to phase separation also seems incongruent with the message of the paper. Further, phase separation of C9orf72 dipeptide repeats in vitro is not, in itself, relevant to disease. Instead, phase separation in vitro has been used to explore interactions that disrupt stress granules, the nucleolus, and nuclear transport, to name a few. It is not clear how PPIA disrupting phase separation of polyPR relates to any proposed disease mechanism of C9orf72. It seems that the authors recognize this fact in that they propose that droplet dissolution is simply a function of PPIA binding to and sequestering polyPR. Measuring droplet dissolution then seems like an unnecessarily convoluted way of getting to the point that polyPR binds to PPIA. I would suggest an experimental strategy that directly addresses how polyPR could tie up PPIA and inhibit native function, as outlined in the authors' model. If data on phase separation is to be included, it needs to be clarified why this is relevant.

My major concern is that most of the data in the manuscript do not seem to address the proposed

mechanism. It is likely beyond the scope of the paper to test these hypotheses in vivo, but some well-designed biochemical assays would go a long way. If the takeaway message is that PPIA will be sequestered on polyPR, I would like to see that polyPR can actually inhibit the native function of PPIA. It is my understanding that the affinity of PPIA to native substrates is substantially higher than the affinity to polyPR.

In summary, I think this is a very clear paper outlining the structure of polyPR bound to PPIA. However, the remaining data that attempts to relate this to toxicity is less convincing. I would like to see the same attention paid to the mechanistic details paid to the structural details.

Reviewer #3 (Remarks to the Author):

Babu et al. report the identification of PPIA as a PR-specific interactor in C9orf72 ALS/FTD. This work follows on an extensive collection of previous papers that have investigated what makes these so-called arginine-rich DPRs, respectively PR and GR, in disease models. While a lot of work has been done in this area, many questions remain unanswered. More specifically, why GR and PR often present with slightly different behaviors and toxicity has been largely neglected. Babu et al. now identify such a potential differential mode of action. Specifically, given the previous implication of PPIA in ALS patients and mouse models, this report provides strong evidence for a role of this protein in the complex pathophysiology of ALS. The presented work is of high quality and the result of carefully designed experiments. It is this reviewers' belief that this paper would be an important addition to the C9orf72 ALS literature as it highlights the molecular details of a novel pathway that may be of therapeutic interest. It is therefore that I can only strongly endorse this paper for publication in Nature Communications, on the condition that the authors to address two comments this reviewer still has.

While I applaud the authors for the high quality structural work presented in this manuscript, given that the authors imply that PR may sequester PPIA into cytoplasmic condensates, it would be good if the authors could test this hypothesis. Is PPIA recruited to PR-induced stress granules? Is this recruited different from canonical stress granules? If the authors could show any evidence that the interaction between PPIA and PR is happening in cells, this would be an invaluable addition to the presented work.

The authors state in the discussion that "the number of PR repeats in C9ORF72-ALS/FTD patients is typically hundreds to thousands". This should be amended, as there is currently no strong data out on the exact size dipeptide repeats in their physiological setting. Especially with the recent reports indicating the existence of frameshifted chimeric DPRs and the fact that PR might inhibit its own translation.

Reviewer #1

The authors present structural evidence of the interaction between a 20-mer repeat of a dipeptide Pro-Arg (PR) and the protein CYP4/PPIA. PR repeats are the result of aberrant protein expression that will lead to neurotoxicity in the brain and contribute to neurodegenerative diseases. Their findings would have implications in finding a molecular mechanism of how a hexanucleotide repeat expansion of the non-coding region of the C9ORF72 gene leads to Amyotrophic Lateral Sclerosis (ALS) and Frontotemporal Dementia (FTD). The manuscript is centered in finding the binding site of the PR 20-mer on the structure of PPIA by means of NMR spectroscopy and X-ray crystallography and provides evidence of binding affinity by NMR data and a droplet disaggregation fluorescence assay. Comparison to a low binding Gly-Arg (GR) 20-mer that belongs to proteins that do not interact with PPIA is provided. The structural evidence presented in the manuscript is sound and provides adequate basis for the conclusions drawn by the authors. The references are appropriate.

Reply: We thank Dr. Russi for evaluating our manuscript and highlighting its importance.

However, major issues were raised: 1. The authors present two residues, Arg55 and Gln111, from their NMR studies (Figure 1b and 1c) as the candidates for binding the 20-mer PR peptide. Whereas Arg55 was confirmed by X-ray diffraction analysis of the complex between the 20-mer and PPIA, no mention of Gln111 is found neither in the main text nor in Figure 2b (which shows hydrogen bonding interactions for Arg55 only). This residue is omitted from Figure 2b. Could the authors explain why this may be happening? Why is the observation from NMR different from X-ray? In fact, Arg55 has been identified in previous X-ray studies of PPIA in complex with Xaa-Pro dipeptides (conf. Zhao and Ke, *Biochemistry* (1996), 35:7362-7368, and PDB structures 2CYH, 3CYH, 4CYH, and 5CYH) where this residue is shown to make hydrogen bonds to the Pro carbonyl while Gln111 is furthest away, not contacting the dipeptide. Several PPIA-bound structures at high resolution have been published that show similar binding interactions. The authors are encouraged to include in their manuscript a reference to these earlier works and compare their new structure to relevant peptide-bound PPIA structures.

Reply: Thanks for pointing it out. Gln111 is not making direct contacts with the PR chain and was therefore not shown. The broadening of the NMR signal of Gln111 during the titration with PR20 is an indirect effect: previous NMR studies have shown that a dynamic network of coupled residues in PPIA is important for enzyme catalysis. In particular, it was shown that the loop formed by residues S110 and Gln111 undergoes chemical exchange and is affected by mutation of Arg55 (reference #35). To stress this point, we state in the revised version of the manuscript (page 4): *“To identify the binding site of PR20 in PPIA, we analyzed the PR20-induced signal broadening for the individual residues of PPIA. The analysis located the strongest intensity decreases within or next to the enzyme’s active site (Fig. 2b,c). The perturbed PPIA residues include Arg55, Gln111, Asn102 and Glu120. Arg55 has been identified in previous X-ray studies of PPIA in complex with Xaa-proline dipeptides where this residue is shown to make hydrogen bonds to the proline carbonyl. Gln111, on the other hand, is not making direct contacts with the PR20 chain (see Fig. 3), but is affected by mutation of Arg55 and is part of a dynamic network of residues in the binding pocket.”*

We also included the structural comparison with the dipeptide complexes as Supplementary Fig. 2.

2. The authors should clarify which parts of the 20-mer are making interactions in describing the packing of PPIA and the 20-mer. In Figure 2d, it is evident that an 8-mer in one asymmetric unit is making interactions with one face of the protein (yellow and light green), and similarly to a symmetry-related molecule (grey). However, it's not evident that there's any interaction with the other packed molecules (orange and sky-blue). In other words, one 8-mer is symmetry-related to a second 8-mer leaving a 4-mer doing a different kind of interactions (or none). Could the authors clarify and expand the discussion on this?

Reply: Please note that always four PR dipeptides bind in an identical manner to the respective PPIA molecule. The impression that there are differences in the interaction mode is just because it is difficult to find a view in which all interactions are equally well seen. To better show that the packed orange molecule does make the same interactions with the PR chain, we introduced an additional figure (new Fig. 3e). In addition, we stress in the manuscript (page 7): *“Always four PR dipeptides from the polymer bind in an identical manner (residues I to I +7 in Fig. 3b) to the respective PPIA molecule in each asymmetric unit (Fig. 3d,e).”*

3. Binding measurements on the peptides and PPIA are based on NMR spectroscopy. The PR peptide shows a low micromolar binding affinity which is ~2+ orders of magnitude stronger than the GR peptide. Did the authors make measurements using a different technique, e.g. ITC, MST, or SPR? These techniques could be rapidly accessible to researchers who want to replicate the experiment (compared to having to produce labelled protein for NMR) and they would provide a direct measurement of binding affinity and stoichiometry. Would the

stoichiometry from crystal packing the same as in solution? Could you explain?

Reply: The PPIA/PR20 affinity was measured by ITC (Fig. 2d). The ITC-derived stoichiometry is somewhere between 1 and 1.5. The NMR binding studies with the shorter PR peptides showed that PR2, the four-residue peptide, does not efficiently bind to PPIA (Supplementary Fig. 1a). In addition, we stress in the discussion section (page 10) that for long PR repeat polymers, which are present in ALS/FTD patients (Fig. 4c), the binding process changes “from a simple one-site binding process to the binding to a one-dimensional lattice with a huge number of potential binding sites (Fig. 4d). Aberrant interactions of C9orf72-repeat polymers are thus mechanistically similar to the binding of transcription factors to DNA in which transcription factors bind to the one-dimensional lattice of binding sites in the DNA.”

4. Minor issues

a. Abstract: GR abbreviation not introduced

b. Pg. 4: "saint score", please add reference: Choi et al., (2011) Nat. Methods 8: 70-73

Reply: We removed the abbreviations from the abstract and instead introduce them now in the manuscript text. The suggested reference was added.

Reviewer #2

The manuscript submitted by Babu et al. sets out to define a novel mechanism of C9orf72 repeat expansion toxicity. In this work, the authors focus on the interaction between polyPR dipeptides and peptidyl proline isomerases (specifically PPIA). The work is motivated by the observation that PPIs have been observed in the interactome of polyPR and not polyGR peptides. Further, knockout of PPIA in mouse models shows similar phenotypes to FTD. Therefore, the authors posit that polyPR toxicity might arise in part by inhibition of PPIA. To test this hypothesis, the authors demonstrate that, among C9orf72 related dipeptide repeats, polyPR has a particularly high affinity and specific interaction with PPIA. The data presented in the manuscript provides a clear picture of polyPR binding to PPIA. This is accomplished via a combination of NMR spectroscopy to map the binding pocket on PPIA and calculate dissociation constant and X-ray crystallography to show the molecular details of the bound structure. Importantly, the authors demonstrate the PR binds substantially tighter than either the proline-free GR or the arginine-free GP dipeptides. The crystal structure outlines a series of interactions involving arginine that explain this specificity. Taken together, I have no issues with the conclusion that polyPR binds specifically to PPIA. From a structural standpoint, these results are novel.

Reply: We thank the referee for evaluating our manuscript and highlighting its importance.

However, the fact that PR (and not other dipeptides) bind to PPIA is perhaps unsurprising given the existing information about C9orf72 interactomes and our knowledge of substrate binding to PPIs. In the interest of clarity, the authors might want to expand on what is meant by slow versus fast exchange when describing the PR versus GP samples. The PR samples look like they are in an intermediate exchange regime due to the lack of peaks associated with the bound state.

Reply: Thanks for the suggestion. We clarify this in the revised version of the manuscript.

I would also like the results from figure 2c/d explained in a bit more detail. It seems to me that the authors are stating that in the crystallization conditions, PPIA binds to polyPR such that one PPI abuts the next. Is this an artifact of crystallization, or is this the presumed functional bound structure?

Reply: We further explain this in the revised version (page 7, now Fig. 3): “The crystal structure of the PR repeat polymer bound to the prolyl isomerase is unique when compared to known protein/protein complex structures (Fig. 3c-e). The chaperone-bound PR polymer chain displays a continuous electron density without interruption throughout the crystal lattice (Fig. 3c-e). However, the electron density of a single asymmetric crystal unit can only accommodate four PR dipeptides. Always four PR dipeptides from the polymer bind in an identical manner (residues I to I +7 in Fig. 3b) to the respective PPIA molecule in each asymmetric unit (Fig. 3d,e). Due to the continuous electron density of the PR chain in the crystal lattice, the PR repeats are randomly positioned, i.e. no unique start of the polymer chain is present (Fig. 3c). This suggests that the chaperone can bind to any of the PR dipeptides in the repeat polymer and the electron density is a result of this averaging process. The continuous electron density thus provides an atomic-resolution view of a large number of chaperone molecules bound to a long PR repeat polymer (Fig. 3d).”

In addition, we stress in the discussion section that “In contrast, disease-associated PR polymers with hundreds to thousands of repeats bind simultaneously many chaperones: based on the crystal structure of the PPIA/PR20 complex a single PR polymer with a repeat size of 2000 can sequester up to 500 prolyl isomerase molecules. In addition, the molecular mechanism changes from a simple one-site binding process to the binding to a one-dimensional lattice with a huge number of potential binding sites (Fig. 4d)⁴³. Aberrant interactions of C9orf72-repeat polymers are thus mechanistically similar to the binding of transcription factors to DNA in which transcription factors bind to the one-dimensional lattice of binding sites in the DNA.”

An additional set of binding experiments that can provide independent validation of the K_d and, ideally, stoichiometry would help these conclusions. Finally, the K_d is calculated as a global fit to four residues in the binding pocket. Does this mean that all 8 residues from polyPR that associate with PPIA bind cooperatively and have the same affinity?

Reply: We performed ITC experiments and added the results to the revised version of the manuscript. We currently don't know if all 8 residues have the same affinity and therefore focus in the revised manuscript on the affinity values derived from Arg55 of PPIA upon titration with PR20, PR10 and PR5 (Fig. 2e). Arg55 was previously shown to be essential for both substrate binding and catalytic activity of PPIA.

The authors additionally characterize the catalytic efficiency of PPIA on polyPR. The NMR assay of exchange is quite interesting. I have no major concerns about this data, but I am intrigued by the apparent slower exchange rate at higher molar ratios. This effect does not seem to be explored in much detail in the text. I would like to see some further explanation for these results. It is also not clear to me where these results fit with the larger story. Seemingly the rate of cis/trans isomerization is unrelated to PR toxicity? The fact that PPIA is a functional proline isomerase for polyPR seems like a red herring. If I am off base on this, the data should be featured in the discussion and related to the rest of the manuscript. The data connecting PPIA to phase separation also seems incongruent with the message of the paper. Further, phase separation of C9orf72 dipeptide repeats in vitro is not, in itself, relevant to disease. Instead, phase separation in vitro has been used to explore interactions that disrupt stress granules, the nucleolus, and nuclear transport, to name a few. It is not clear how PPIA disrupting phase separation of polyPR relates to any proposed disease mechanism of C9orf72. It seems that the authors recognize this fact in that they propose that droplet dissolution is simply a function of PPIA binding to and sequestering polyPR. Measuring droplet dissolution then seems like an unnecessarily convoluted way of getting to the point that polyPR binds to PPIA. I would suggest an experimental strategy that directly addresses how polyPR could tie up PPIA and inhibit native function, as outlined in the authors' model. If data on phase separation is to be included, it needs to be clarified why this is relevant. My major concern is that most of the data in the manuscript do not seem to address the proposed mechanism. It is likely beyond the scope of the paper to test these hypotheses in vivo, but some well-designed biochemical assays would go a long way. If the takeaway message is that PPIA will be sequestered on polyPR, I would like to see that polyPR can actually inhibit the native function of PPIA. It is my understanding that the affinity of PPIA to native substrates is substantially higher than the affinity to polyPR. In summary, I think this is a very clear paper outlining the structure of polyPR bound to PPIA. However, the remaining data that attempts to relate this to toxicity is less convincing. I would like to see the same attention paid to the mechanistic details paid to the structural details.

Reply: We agree with the reviewer that the isomerization and LLPS assays are not directly related to the proposed mechanism, that is inhibition of PPIA-catalyzed protein folding by PR repeat polymers (Fig. 4). We therefore removed the two assay data from the manuscript and instead included new data, which demonstrate that PR repeat polymers directly inhibit PPIA-catalyzed protein folding (new Fig. 1b,c; pages 3-4, 8). To reflect the new data, we also modified the introduction and discussion sections.

Reviewer #3

Babu et al. report the identification of PPIA as a PR-specific interactor in C9orf72 ALS/FTD. This work follows on an extensive collection of previous papers that have investigated what makes these so-called arginine-rich DPRs, respectively PR and GR, in disease models. While a lot of work has been done in this area, many questions remain unanswered. More specifically, why GR and PR often present with slightly different behaviors and toxicity has been largely neglected. Babu et al. now identify such a potential differential mode of action. Specifically, given the previous implication of PPIA in ALS patients and mouse models, this report provides strong evidence for a role of this protein in the complex pathophysiology of ALS. The presented work is of high quality and the result of carefully designed experiments. It is this reviewers' belief that this paper would be an important addition to the C9orf72 ALS literature as it highlights the molecular details of a novel pathway that may be of therapeutic interest.

It is therefore that I can only strongly endorse this paper for publication in Nature Communications, on the condition that the authors to address two comments this reviewer still has.

Reply: We thank the referee for evaluating our manuscript and highlighting its importance.

While I applaud the authors for the high quality structural work presented in this manuscript, given that the authors imply that PR may sequester PPIA into cytoplasmic condensates, it would be good if the authors could test this hypothesis. Is PPIA recruited to PR-induced stress granules? Is this recruited different from canonical stress granules? If the authors could show any evidence that the interaction between PPIA and PR is happening in cells, this would be an invaluable addition to the presented work.

Reply: Please note that the in cell interaction between PPIA and PR has been demonstrated by three independent proteomics studies (please see references #5,6,7). However to reliably identify in which cellular condensates the interaction occurs is beyond the scope of the current manuscript. As suggested by reviewer #2 we replaced the phase separation data by new experimental data demonstrating that PR repeat polymers inhibit the folding catalyst activity of PPIA (Fig. 1). The new data are described in the results section (pages 4-5) and in the discussion and provide strong support for the proposed mechanism, i.e. that PR repeat polymers inhibit PPIA-catalyzed protein folding and thus perturb protein homeostasis in C9ORF72-ALS/FTD (Fig. 4).

The authors state in the discussion that “the number of PR repeats in C9ORF72-ALS/FTD patients is typically hundreds to thousands”. This should be amended, as there is currently no strong data out on the exact size dipeptide repeats in their physiological setting. Especially with the recent reports indicating the existence of frameshifted chimeric DPRs and the fact that PR might inhibit its own translation.

Reply: We now show data about repeat lengths in C9ORF72 patients enlisted in the Answer ALS project (Fig. 4c) and refer to this figure when making the statement. For the enlisted individuals, persons classified as healthy have less than 30 repeats, while diseased individuals have at least hundreds of repeats.

REVIEWER COMMENTS

Reviewer #1 (Remarks to the Author):

The authors addressed all the points and no further questions arose. I recommend the publication of the manuscript with the revisions that have been made.

Reviewer #2 (Remarks to the Author):

see attached

Reviewer #3 (Remarks to the Author):

As stated in my original comments, I deemed this manuscript suitable for publication if the authors could address two of my issues with the manuscript as presented. Disappointingly, the authors have not addressed any of these comments.

The first comment pertains to the interaction between PPIA and PR in cells. The authors have removed the phase separation data from the manuscript, which indeed was confusing as it did not add any meaningful biological insight as pointed out as well by other reviewers. Yet, the reader is now still left with the question if this interaction happens in the cellular context? The authors point at three different mass spec studies suggesting that the interaction happens in cells. One of these studies used pulldowns from cell lysate, while the two other studies used precipitation of cellular protein complexes and nucleic acids by PR-induced coacervation. This is definitely useful information, yet, the reader is left wondering if and where these interactions happen in the cell, especially since GR, PR and GP all show different subcellular localization patterns. The authors have undoubtedly shown that PR can bind to and interfere with the activity of PPIA in a test tube, what this means for a cell remains to be seen.

The second comment addressed a misrepresentation of the literature. In the initial version the authors suggested that the PR repeats are hundreds or thousands units in length. Instead of amending this false statement, as suggested, the authors decided to double down on misrepresenting the literature and even go a step further:

"However, PR/chaperone interactions with low stoichiometry only occur in healthy people, which always have short dipeptide repeat polymers (< ~30 repeats)⁴¹."

The authors make two key mistakes here:

(1) They assume the DNA repeat length equals peptide repeat length. There is no evidence whatsoever to indicate this. We know that RAN translation can occur anywhere without the need of an initiating start codon, there is translational stalling by PR repeats, we know about translational frameshifting on the C9 repeats, and there is abortive transcription of the repeat itself leading to the formation of short repeat transcripts. All these mechanisms will result in a large pool of repeat peptide species that are shorter than what is theoretically possible. If the authors claim that these short peptides would be pathologically meaningless, then please explain why numerous disease models (e.g., fly, zebrafish, yeast, cell lines, iPSC/primary neurons,...) show dramatic toxicity when using repeats of this size range. The authors are free to support any hypothesis regarding the physiological length of PR peptides in patients, but they will just need to provide actual experimental evidence rather than formulating base-less claims.

(2) The authors make another unfounded claim that non-C9 carriers have dipeptide repeats that are smaller than 30 repeat units. There is no experimental evidence whatsoever to claim that DPRs are formed in the absence of expanded repeat transcripts. The definition of RAN translation involves the

need of expanded repeats to drive their recognition by the translational machinery. While the exact mechanism of RAN translation and the precise repeat size cut-off remains unknown, stating that non-carriers produce short peptide repeats would be a major scientific finding that needs to be supported by actual experimental evidence, which is not found in the review that the authors cite.

It is very disappointing to see the authors try to misrepresent the current scientific evidence regarding the true size of DPRs in patients, instead of just amending their initial false statement and provide the necessary nuance. Yes, most likely C9 patients present with a heterogeneous pool of peptides of different lengths (including large ones) and involving chimeras, yet, stating that non-carriers produce short peptides and carriers produce only large ones is factually incorrect.

After carefully assessing the revised manuscript, I still stand by my initial evaluation of this paper. The authors provide a wonderful and in-depth characterization of the inhibitory effect of PR on PPIA in the test tube. Yet, whether this mechanism is at play within cells remains to be seen. Additionally, the misrepresentation of the C9 and RAN translation literature regarding DPR lengths needs to be corrected.

Reviewer #2 Attachment:

Babu et al. Revision

The revisions have satisfied most of my major concerns with the original manuscript. Importantly, I applaud the authors on the improved clarity and shifted focus onto the novel structural insights. I stand by my initial assessment that the structural data does add an exciting element to the ongoing research in the c9orf72 DPR field.

I have several minor comments.

1. The protein folding assays presented in figure 1 help convey the point that polyPR peptides can inhibit the native function of PPIA. However, the assays presented show a minimum 7x excess in the ratio of PPIA and polyPR. It would be helpful to compare these ratios and what is expected when DPRs are translated. If this is simply competitive inhibition, the authors could use the measured affinity of polyPR for PPIA to model this behavior. I suspect that the authors' point is that pathological DPRs contain many more repeats than the PR20 tested. Is some avidity expected with longer DPRs that would increase the inhibition of PPIA?
2. Given the use of the term "exponential constant," I assume this data is fit to a single exponential? Please clarify this in the methods. If these fits are shown, I cannot make them out in my printed version of the figures. I'm a touch surprised that a single exponential and not a double is all that is required to fit this data.

Minor point: The data overlays in panel b such that all traces are not visible.

As a final suggestion, I believe the authors could improve on the discussion of how this potential mechanism of pathology fits with the existing canon of suggested impacts of long DPR translation. I'm curious if there are other impacts of inhibiting proline isomerase activity that could antagonize or synergize with existing proposed mechanisms of toxicity such as those listed by the authors in the discussion.

Reviewer #2

The revisions have satisfied most of my major concerns with the original manuscript. Importantly, I applaud the authors on the improved clarity and shifted focus onto the novel structural insights. I stand by my initial assessment that the structural data does add an exciting element to the ongoing research in the c9orf72 DPR field.

Reply: We thank the referee for having another look at our manuscript and highlighting its importance.

I have several minor comments.

1. The protein folding assays presented in figure 1 help convey the point that polyPR peptides can inhibit the native function of PPIA. However, the assays presented show a minimum 7x excess in the ratio of PPIA and polyPR. It would be helpful to compare these ratios and what is expected when DPRs are translated. If this is simply competitive inhibition, the authors could use the measured affinity of polyPR for PPIA to model this behavior. I suspect that the authors' point is that pathological DPRs contain many more repeats than the PR20 tested. Is some avidity expected with longer DPRs that would increase the inhibition of PPIA?

Reply: Thanks for these thoughts. We indeed think that it is likely that in pathological DPRs avidity might result in increased inhibition of PPIA. Avidity might also involve direct interactions between PPIA molecules, when these molecules are bound to proximal PR sites in the DPR chain. Another complication is that we have no estimates for the concentrations of PPIA in proximity to the ribosome, i.e. the molar ratios of PPIA and polyPR when polyPR is translated. We therefore currently don't feel confident to model the binding of multiple PPIA molecules to the effectively one-dimensional lattice of the polyPR chain. However, we added a sentence to the discussion section stating the potential contribution of avidity to the inhibition of PPIA by DPRs (page 10).

Given the use of the term "exponential constant," I assume this data is fit to a single exponential? Please clarify this in the methods. If these fits are shown, I cannot make them out in my printed version of the figures. I'm a touch surprised that a single exponential and not a double is all that is required to fit this data. Minor point: The data overlays in panel b such that all traces are not visible.

Reply: Yes, these are mono-exponential fits (now stated in the methods). All individual refolding curves and their fits are now shown in the new SI Fig. 1. In some cases, a biphasic fit is better (e.g. for RNaseT1:PPIA=1:0.033 ; SI Fig. 1). However, k2 values which were derived through biphasic fitting for the second phase of the folding curve had large errors. We therefore restricted the analysis to mono-exponential fitting for all conditions.

As a final suggestion, I believe the authors could improve on the discussion of how this potential mechanism of pathology fits with the existing canon of suggested impacts of long DPR translation. I'm curious if there are other impacts of inhibiting proline isomerase activity that could antagonize or synergize with existing proposed mechanisms of toxicity such as those listed by the authors in the discussion.

Reply: Indeed there might be other impacts of binding of polyPR to the active site of PPIA, because PPIA and other proline isomerases not only promote cis/trans-isomerization but also can bind to hydrophobic sequences, which do not contain proline residues. This binding could affect a number of potential toxic mechanisms associated with polyPRs (such as perturbation of membrane-less organelles by polyPRs or perturbed nucleocytoplasmic transport). However, we think it is better to keep the focus on the impact on protein folding, in line with the data presented in the manuscript.

Reviewer #3

As stated in my original comments, I deemed this manuscript suitable for publication if the authors could address two of my issues with the manuscript as presented. Disappointingly, the authors have not addressed any of these comments. The first comment pertains to the interaction between PPIA and PR in cells. The authors have removed the phase separation data from the manuscript, which indeed was confusing as it did not add any meaningful biological insight as pointed out as well by other reviewers. Yet, the reader is now still left with the question if this interaction happens in the cellular context? The authors point at three different mass spec studies suggesting that the interaction happens in cells. One of these studies used pulldowns from cell lysate, while the two other studies used precipitation of cellular protein complexes and nucleic acids by PR-induced coacervation. This is definitely useful information, yet, the reader is left wondering if and where these interactions happen in the cell, especially since GR, PR and GP all show different subcellular localization patterns. The authors have

undoubtedly shown that PR can bind to and interfere with the activity of PPIA in a test tube, what this means for a cell remains to be seen.

Reply: We thank the referee for further evaluating our manuscript. While we feel that the three different mass spec studies convincingly demonstrate that PR dipeptide repeats interact with PPIA in a cellular context, we agree that we currently do not know where in the cell these interactions occur. We therefore stress in the discussion section that our study further warrants studies to investigate the interaction of PR dipeptide repeats with PPIA in neurons including their subcellular localization (page 10).

The second comment addressed a misrepresentation of the literature. In the initial version the authors suggested that the PR repeats are hundreds or thousands units in length. Instead of amending this false statement, as suggested, the authors decided to double down on misrepresenting the literature and even go a step further: "However, PR/chaperone interactions with low stoichiometry only occur in healthy people, which always have short dipeptide repeat polymers (< ~30 repeats)⁴¹." The authors make two key mistakes here: (1) They assume the DNA repeat length equals peptide repeat length. There is no evidence whatsoever to indicate this. (2) The authors make another unfounded claim that non-C9 carriers have dipeptide repeats that are smaller than 30 repeat units... It is very disappointing to see the authors try to misrepresent the current scientific evidence regarding the true size of DPRs in patients, instead of just amending their initial false statement and provide the necessary nuance. Yes, most likely C9 patients present with a heterogeneous pool of peptides of different lengths (including large ones) and involving chimeras, yet, stating that non-carriers produce short peptides and carriers produce only large ones is factually incorrect. After carefully assessing the revised manuscript, I still stand by my initial evaluation of this paper. The authors provide a wonderful and in-depth characterization of the inhibitory effect of PR on PPIA in the test tube. Yet, whether this mechanism is at play within cells remains to be seen. Additionally, the misrepresentation of the C9 and RAN translation literature regarding DPR lengths needs to be corrected.

Reply: We thank the referee for the detailed explanation. We agree that further experimental evidence is needed to define the length of DPR proteins in patients and the connection between DPR protein length and toxicity/disease severity. We therefore removed the previous Fig. 4c as well as the sentence "However, PR/chaperone interactions with low stoichiometry only occur in healthy people" In the revised version of the manuscript, there is no more statement about DPR length in C9orf72 patients.